# The Impact of Green Innovation on Corporate Performance: An Analysis Based on Substantive and Strategic Green Innovations

**Mingxia Liu [1], Liqian Liu [1,\*] and Amei Feng [2]**

1    School of Economics and Management, Wuhan University, Wuhan 430072, China; liumingxia@whu.edu.cn
2    School of Economics, Wuhan Textile University, Wuhan 430072, China; ameifeng001@163.com
\*    Correspondence: liuqian210@whu.edu.cn

**Abstract:** Green innovation is a new approach to achieving sustainable social development. Examining whether firms can reap the rewards of this costly and risky endeavor is essential to assessing whether they can sustainably adhere to a green strategy. This study was conducted on a sample of Chinese A-share-listed firms from 2010 to 2021 and employed a two-way fixed-effects approach. We found that substantive and strategic green innovations significantly impact firms' financial and environmental performance. Specifically, substantive green innovation leads to a significant improvement in financial performance, while strategic green innovation weakens financial performance; both types of green innovations lead to a significant improvement in environmental performance, with strategic green innovation being more effective in this regard compared to substantive green innovation. Moreover, our heterogeneity analyses showed that substantive green innovation has a weaker effect on improving financial performance in state-owned enterprises (SOEs) and in firms in regions with higher government environmental concerns; similarly, in SOEs, strategic green innovation has a weaker detrimental effect on financial performance. The findings of this study provide substantial evidence for promoting green innovation transformation and the upgrading of enterprises.

**Keywords:** green innovation; substantive green innovation; strategic green innovation; financial performance; environmental performance

## 1. Introduction

Environmental crises such as degradation and resource scarcity have become increasingly severe in recent years, making pollution control and sustainable resource use critical global challenges. The introduction of sustainable development has made environmental protection an integral part of economic development for countries worldwide, including China. Scholars believe that green technology development and diffusion are fundamental to improving energy efficiency, achieving energy savings, reducing emissions, and realizing sustainable economic and environmental development [1]. Firms engage in green innovation not only to respond to environmental and social challenges but also to achieve long-term sustainable economic growth [2], especially in a rapidly developing economy such as China [3]. In light of this, the Chinese government has issued several policy documents on green innovation to encourage companies to pursue green and sustainable development paths, guiding economic growth from high speed to high quality.

Compared to traditional innovation, green innovation faces a double externality problem [4]. The conventional view is that environmental regulations are an expensive burden that governments impose on firms. Firms must invest in unproductive activities to reduce pollution to the environment, which reduces their profitability [5]. However, Porter and Vanderlinde argue that appropriately designed environmental standards can stimulate firms to innovate, offset compliance costs, and enhance their competitiveness [6]. This is known as the "Porter hypothesis". Yu et al., argued that although environmental regulations can cause firms to incur higher costs initially, they can still respond effectively to

external environmental pressures by implementing green innovation strategies [7]. And, in the long run, such strategies can also help firms to significantly improve their sustainable competitive advantage and financial performance [7,8]. Despite the relatively strong theoretical and empirical evidence on the environmental benefits of green innovation [9–12], there is an ongoing debate about whether it can improve firms' financial performance [13,14].

The existing research on the relationship between green innovation and firm performance has several limitations. First, a large portion of the studies is based either on cross-sectional data obtained from questionnaires [15–19] or using panel data but only based on the consideration of the composite construction of "green innovation" [20–22], resulting in inconsistent findings between green innovation and corporate financial performance [14,23]. Second, most studies focus only on the single impact of green innovation on firms' financial performance [1,24,25] or on firms' comprehensive performance indicators [26,27], and the literature lacks comparative studies on the performance consequences of different dimensions of green innovation (e.g., financial performance and environmental performance) [23]. Green innovation is a complex of corporate efficiency and legitimacy. The conclusions drawn from focusing only on financial performance indicators are biased and may not be sufficient to fully capture the impact of green innovation on firm performance, especially in terms of long-term and environmental performance [3].

In this context, scholars argue that the benefits of green innovation to firms are related to the quality and type of innovation [28–30]. According to the type of green innovation content, Chen and Liu [31] argue that there is a significant difference between the impacts of green product innovation and green process innovation on firms' financial performance. However, more evidence is needed. Under the pressure of regulatory policies and the induction of incentive policies, some firms engage in "greenwashing" behavior (i.e., developing low-quality strategic green innovation) while avoiding the development of high-input, high-risk, and high-quality substantive green innovation [32–34]. According to a study by Zhang et al. [3], the performance growth of Chinese state-owned enterprises (SOEs) is mainly driven by low-quality green patents. However, another study conducted by Zhao et al. [35] confirms that high-quality green innovations are essential in achieving a win–win situation for both the economy and the environment under environmental regulation. Further research is required to determine whether there is a difference in the effects of high-quality substantive green innovation and low-quality strategic green innovation on a firm's performance, or if the "Porter hypothesis" holds for both types of green innovations.

Green innovation encompasses both green technological innovations, which involve new or improved products and processes, and non-green technological innovations, such as new management or business models that improve environmental sustainability [4,13]. This study focuses explicitly on green technological innovation. In previous studies, green innovation has often been used to refer to green technological innovation [10,20], and, therefore, this study also uses the term green innovation to refer to green technological innovation. Scholars state that firms are made up of several components or dimensions and that assessing a firm based on a single dimension is flawed [36]. Baah et al., also point out a need to integrate financial and non-financial dimensions in assessing corporate performance [36]. Based on this recommendation and considering the improvement of the environment as the original purpose of green innovation, we, therefore, employ both financial (financial performance) and non-financial indicators (environmental performance) in assessing firm performance, which aligns with existing studies [3,36,37].

Based on stakeholder theory, this study aims to capture the characteristics of green innovations that are vital contributors to different dimensions of firm performance by examining whether substantive green innovation and strategic green innovation have different impacts on corporate financial and environmental performance. Furthermore, this study investigates the heterogeneity of this influential relationship in the face of different firm ownerships and different levels of regional government environmental concern.

The contributions of this study are as follows. First, this study analyzes the impact of green innovation on corporate financial and environmental performance over 12 years using a two-way fixed-effects model with panel data from listed companies in China. This extends previous studies of cross-sectional data and adds to the green innovation literature. Second, the findings reveal that the contributions of substantive green innovation and strategic green innovation to different dimensions of firm performance are significantly different. Substantive green innovation can achieve benefits in both financial and environmental performance, while strategic green innovation obtains better environmental outcomes at the expense of financial performance. These findings have important implications for distinguishing between different types of green innovations, both theoretically and operationally. Third, this study shows that firms' internal and external characteristics, such as ownership and government environmental concerns, affect the relationship between substantive green innovation and financial performance. This finding provides a new perspective for understanding the complex link between firms' green innovation and performance and enriches contextual studies on the relationship between green innovation and firm performance. Fourth, this study provides a corresponding reference basis for the formulation of green innovation incentive policies in emerging economies represented by China and can enable corporate managers to make reasonable investment decisions in environmental protection by understanding the relationship between green innovation and firm performance in different quality dimensions.

The paper is structured as follows: Section 2 presents the literature review and theoretical hypotheses. Section 3 outlines data processing and model setting. The empirical results, robustness test, and heterogeneity test of this study are presented and discussed in Section 4. Then, the conclusion, insights, and limitations are provided in Section 5.

## 2. Literature Review and Theoretical Hypothesis

### 2.1. Literature Review

#### 2.1.1. Green Innovation

When referring to green innovation, different terms are used in existing research, such as "eco-innovation", "environmental innovation", and "sustainable innovation". Although green innovation has been defined from various perspectives by many scholars and institutions, all emphasize the importance of maximizing the use of natural resources while minimizing negative impacts on the environment. According to the type of innovation content, green technological innovation is further categorized into green product innovation, which generates environmental benefits for end-users when they use the product or service, and green process innovation, which generates environmental benefits for firms in the process of producing the product or service [10]. Recent studies classify green innovations into high-quality substantive green innovation and low-quality strategic green innovation based on the motives behind innovation [33,38]. Existing studies point out that green product innovation that eliminates the generation of pollutants at the source is the highest level of green technological innovation [10]. Furthermore, green process innovations generally encompass integrated cleaner production technologies that reduce pollution generation during the production process and end-of-pipe solutions that treat pollution after it has been generated. The former is considered a higher level of technological content among green innovation, while the latter is classified as a lower level of technological content [39].

#### 2.1.2. The Impact of Green Innovation on Corporate Performance

Green innovation and financial performance: Several studies have shown a positive relationship between green innovation and firms' financial performance. For example, Li et al., argued that manufacturing firms can generate new profit growth by commercializing green technologies and generate additional revenue by converting recyclable waste into marketable goods [16]. According to Afum et al. [40], implementing green manufacturing helped firms improve their financial performance by redesigning their production systems to comply with strict environmental laws and regulations and gain a competitive advan-

tage by producing high-quality products. Furthermore, green technology innovations can improve product quality and adaptability and reduce operating costs, which tends to improve the performance of manufacturing firms [26]. According to Hu et al. [41], by distinguishing between green process innovation and green product innovation, they explored how government green subsidies indirectly affect the financial performance of firms through these two types of innovations and found that by fully utilizing green process and product innovations, firms can become more competitive and sustainable, as well as improve financial performance. However, some studies have shown that green innovation practices have a negative impact on firms' current financial performance due to the significant additional costs associated with implementing green innovation. For example, a study suggested that focusing too much on green innovation relative to other innovations hurts accounting and stock market performance [22]. Interestingly, some studies have shown a non-linear correlation between green innovation and financial performance. For instance, using five years of data from more than 6000 French firms, Pekovic et al., found that the impact of environmental investments on economic performance follows an almost inverted U-shaped curve [5]. Additionally, some scholars argue that green innovation does not impact firms' financial performance. For example, a study found no improvement in the financial performance of firms implementing green innovation compared to non-green innovating firms, i.e., firms that engage in green innovation do not necessarily reap financial benefits [42]. In addition, it has been argued that the economic returns that green innovations generate for firms depend on the type of green innovation. According to Rexhäuser and Rammer [30], green innovations can be categorized into regulatory-induced and voluntary. They observed that innovations that do not improve the resource efficiency of firms do not provide positive profitability returns. However, innovations that enhance firms' resource efficiency positively impact material or energy consumption profitability per unit of output.

Green innovation and environmental performance: Firms' environmental performance is viewed as their ability to reduce the generation and emission of pollution and improve energy efficiency, including reducing the generation and emission of wastewater, exhaust, and solid waste, reducing the consumption of natural resources, and reducing the frequency of environmental accidents. Several empirical studies have examined the relationship between green innovation and environmental performance and found that green innovation can significantly contribute to it. For instance, the study by Roh et al. [12] found that green innovation can enhance a company's environmental performance through measures such as energy conservation, emission reduction, and minimizing the use of hazardous substances. The research by Yu et al. [7] stresses the significance of green innovation in enhancing companies' environmental sustainability. It emphasizes that firms can effectively address external environmental pressures by adopting green innovation strategies, which can ultimately lead to better environmental performance. In addition, in terms of the types of content of green innovation, green process innovation helped firms reduce cleaner production costs and lower pollutant emissions to comply with environmental regulations [43]. Similarly, green product innovation enabled firms to reduce resource wastage and reduce the negative environmental impacts of products during consumer use [37]. Previous research has also found that, by developing green process innovation and product innovation activities, firms may fundamentally change how they operate their existing products and processes and may even create new products and processes with significantly reduced negative environmental impacts, resulting in significant improvements to environmental performance [9].

## 2.2. Theoretical Hypothesis

### 2.2.1. Green Innovation and Financial Performance

In the current business landscape, green innovation has emerged as a crucial strategy for companies looking to gain a competitive advantage [44,45]. By leveraging eco-friendly solutions to produce superior and innovative products, organizations can reap numerous

benefits, including enhancing their brand reputation, garnering support from stakeholders, expanding their market share, and boosting customer satisfaction [18,46].

Based on stakeholder theory, firms that prioritize environmental responsibility can better manage their relationships with stakeholders and gain their support [8,47], which ultimately affects their performance. This is because stakeholders are increasingly concerned about the performance of firms in relation to environmental responsibility [48]. It is worth noting that high-quality substantive green innovation and low-quality strategic green innovation differ in terms of meeting the demands of different stakeholders, which in turn can lead to different impacts on a firm's profitability.

Integrating stakeholders' environmental perspectives into a firm's value chain through green innovation may affect the firm's costs and revenues. On one hand, enterprises can increase revenue by developing green products [41]. Since firms that achieve green innovation have the legitimacy to charge higher prices to consumers [44], by incorporating environmental issues into new products, firms can enhance product differentiation, which leads to increased recognition by organizational stakeholders and eventually results in higher competitiveness and sales [17,20,49]. The government's increasing focus on environmental regulations and the consumers' growing environmental awareness provides an opportunity for firms to stand out by offering differentiated green products. This helps them gain a first-mover advantage, making their products marketable and increasing their competitiveness [10]. On the other hand, green process innovation can also affect firms' costs. By adopting cleaner production technologies with a higher technology level, firms can not only improve the efficiency of energy and material use per unit but also reduce resource wastage by increasing the recycling of waste materials, thereby lowering business costs [41]. On the contrary, using end-of-pipe solutions with a lower technology level does not affect the production and processes of firms but instead increases equipment acquisition and operating costs [42,50].

Only when the quality of green innovation is sufficiently high can firms improve their competitiveness while complying with environmental regulations and also realize the compensatory effects of innovation under environmental regulation [35]. Implementing substantive green innovation with high quality may lead firms to redesign their production processes and services, which plays a vital role in developing new environmentally friendly materials and improving energy efficiency. Redesigning production processes can lead to synergies between different resources and capabilities by regrouping firms' resources, thereby increasing their profitability. Furthermore, substantive green product innovation with high technological content throughout the value chain or life cycle can create competitive advantages by strategically capturing markets or establishing standards in their favor by meeting consumers' environmental aspirations [1].

Strategic green innovations, such as specific filters and treatment technologies distributed to end-of-pipe technologies, may help reduce pollution, but they increase business costs [51]. These innovations are developed to help firms comply with relevant environmental policies and regulations but do not fundamentally change the production process. Strategic green innovation fails to improve firms' profitability because they cannot optimize their resources and capabilities. In fact, they burden firms with significant investments and reduce their financial flexibility, ultimately resulting in a loss in financial performance. To summarize, strategic green innovation is an unproductive activity that may help firms meet regulatory standards but can hurt their profitability. Therefore, we propose the following hypotheses:

**H1a.** *Substantive green innovation has a positive impact on corporate financial performance.*

**H1b.** *Strategic green innovation has a negative impact on corporate financial performance.*

### 2.2.2. Green Innovation and Environmental Performance

As the world becomes increasingly aware of the impact that businesses have on the environment, it is essential that companies take action to address the environmental demands and expectations of stakeholders. Firms can respond to stakeholders' environmental demands and claims by adopting green technology innovation. This, in turn, can help firms improve their environmental performance by reducing their pollutant emissions and minimizing the negative impact on the environment.

We argue that any type of green technology innovation can improve the environmental performance of a firm to some extent [17]. First, green process innovation can help firms reduce the generation and emission of pollutants, thus minimizing environmental accidents and their negative impact on the environment [40]. Firms that meet or even exceed the requirements of relevant environmental laws and regulations can also gain legitimacy by avoiding penalties and gaining approval from regulatory stakeholders such as governments [52].

Second, firms can also respond to the growing market and consumer demand for environmentally friendly products by developing products that are sustainable and eco-friendly. By considering the green attributes of their products throughout the product life cycle, firms can optimize environmental benefits in the dimensions of product design, consumer use and maintenance, and recycling [43].

Third, green innovation practices can also help businesses improve their overall environmental impact by developing new materials that reduce waste and dependence on non-renewable resources. By reducing the frequency of three-waste emissions and environmental accidents, companies can make a significant contribution to the preservation of our environment [40,53]. Based on this, we propose the following hypotheses:

**H2a.** *Substantive green innovation has a positive impact on corporate environmental performance.*

**H2b.** *Strategic green innovation has a positive impact on corporate environmental performance.*

Previous research has pointed out that regulatory pressure is one of the main drivers of green innovation [14]. Environmental regulations force firms to invest in green innovations to reduce pollution and avoid negative environmental impacts. However, this regulatory pressure drives mainly strategic green innovation rather than substantive green innovation [54]. Companies that are driven by regulatory pressure engage in green innovation not to achieve commercial success but rather to meet emission reduction targets and gain legitimacy in environmental terms through green innovation.

Strategic green innovations that focus more intensively on end-of-pipe abatement technologies have a more significant impact on reducing pollution and meeting environmental standards than substantive green innovations. Developing strategic green innovations, such as pollution interception and treatment technologies in end-of-pipe treatment programs, helps firms comply with relevant environmental policies and regulations and reduces waste emissions directly and significantly, which has a more direct overall impact on environmental performance. Meanwhile, substantive green innovation has a less immediate and intuitive environmental impact than strategic green innovation. Substantive green innovations, such as substantive green product innovation and substantive clean process innovation, indirectly impact the environment and take longer to be realized [38]. Therefore, strategic green innovation is more advantageous in terms of firms' environmental performance and in responding to regulatory pressures from stakeholders. Based on this, we propose the following hypothesis:

**H2c.** *Strategic green innovation contributes more to corporate environmental performance than substantive green innovation.*

### 3. Research Design

*3.1. Sample Selection*

For our research, we used China's A-share-listed companies as a sample. We set the sample interval as 2010–2021 based on the availability of green patents, carbon emissions, and other related data. After collecting the data in the sample interval, we processed the raw data as follows: (1) financial and insurance companies were excluded; (2) companies with irregular trading (ST and *ST enterprises) were excluded; (3) samples with missing observations of core variables (green patent licensing and carbon emission data) were excluded; (4) all continuous variables were shrink-tailed by 1% up and down (Winsorized) to prevent extreme values. Then, we obtained an unbalanced panel of 10,940 valid observations over 12 years, consisting of 2373 firms that are all Chinese A-share-listed companies and have implemented green innovations, either substantive or strategic or both. Regarding regional distribution, there are 7830 observations from 1725 eastern companies, 1822 observations from 390 central companies, and 1287 observations from 258 western companies. In terms of industry distribution, there are 9323 observations from 2072 manufacturing companies, 521 observations from 98 construction companies, 349 observations from 81 electricity and heat production and supply companies, and 747 observations from 122 companies in 11 other industries. Companies listed on China's A-share market are generally of a certain size, with the mean number of employees in our sample being 8735 and the median being 2900.

The data sources for this paper consisted of three main parts: First, green innovation-related data were obtained from the China Research Data Service Platform (CNRDS). Second, corporate carbon emissions data were obtained from annual reports of the listed companies, which were manually collated. Third, the data on corporate finance and other firm characteristics were sourced from CSMAR.

*3.2. Definition of Variables*

3.2.1. Dependent Variables

The dependent variables were financial performance (FP) and environmental performance (EP). Referring to the existing research [20,25,35,42,55], we used the return on assets (ROA) in the CSMAR database to measure corporate financial performance (FP). In reference to existing studies [37,51], we chose firms' carbon emissions to measure environmental performance. Specifically, we used carbon dioxide ($CO_2$) emissions per unit of assets to measure corporate environmental performance (EP). This is an inverse indicator, with lower values indicating lower carbon emissions and better environmental performance. We used ROE and carbon dioxide emissions per capita as alternative measures of financial performance (FP1) and environmental performance (EP1) in the robustness tests, respectively.

3.2.2. Independent Variables

The independent variables were substantive green innovation (LnSubGI) and strategic green innovation (LnStrGI). Patents are a commonly used indicator to measure the outcome of research and development activities and the protection of industrial property rights [56]. In line with existing studies [3,33,57], we measured substantive and strategic green innovations through the number of green patents granted by firms. Precisely, substantive green innovation (LnSubGI) was measured by the number of green invention patents granted by firms, and the number of green utility model patents granted was used to measure strategic green innovation (LnStrGI). The logarithms of these two variables were used in this paper. Additionally, we provide descriptive statistics of the non-logarithmic treatment of the two green patent grant numbers, SubGI and StrGI, and the total green patent grant numbers, TotalGI.

3.2.3. Control Variables

This study included a set of variables for controlling the potential effects of other factors on the relationship between green innovations and financial and environmental performance. Previous studies have shown that firm size [16,25] and government subsidy [58] significantly moderate the relationship between green innovation and firms'

financial performance. In addition, previous studies have also found that ISO14001 [1,24] is a critical variable in predicting firms' environmental performance. In addition, control variables mentioned in previous studies included the firm fixed asset ratio [3], research and development investment [20,59], and gearing ratio [1,59]. Therefore, our model contains the following control variables: (i) enterprise size (Size): the logarithm of the total assets of the enterprise at the end of the year; (ii) fixed asset ratio (Fixed): the proportion of fixed assets to total assets; (iii) gearing ratio (Lev): the proportion of total liabilities to total assets; (iv) research and development investment (R&D): the proportion of R&D investment to the operating revenue; (v) government subsidies (Subsidy): the logarithm of the amount of government subsidies received by the enterprise in the year; (vi) ISO14001 certification (ISO14001): if the enterprise passes the ISO14001 audit in the year, the value is assigned as 1, otherwise it is 0. Since green patents are already in use and have begun to have an impact on corporate performance during the application process, we chose the current year period for all variables. The specific definitions of the variables are shown in Table 1. In addition, we controlled for firm (Individual) and year (Year) for all models.

**Table 1.** Variables definition.

| Category | Variable Name | Measurement | Reference | Data Source |
|---|---|---|---|---|
| Dependent variables | Financial Performance (FP) | The return on total assets (ROA). | Aguilera-Caracuel and Ortiz-de-Mandojana (2013) [42]; Akbar et al., (2021) [55]; Rezende et al., (2019) [20]; Zhao et al., (2022) [35] | CSMAR |
| | Environmental Performance (EP) | Carbon dioxide emissions per unit of assets. | Fujii et al., (2013) [51]; Lee and Min (2015) [37] | Annual reports of enterprises |
| Independent variables | Substantive Green Innovation (LnSubGI) | The natural logarithm of the number of green invention patents authorized. | Liao (2020) [57]; Jiang and Bai (2022) [33] | CNRDS |
| | Strategic Green Innovation (LnStrGI) | The natural logarithm of the number of green utility model patents authorized. | | CNRDS |
| Control variables | Enterprise size (Size) | The logarithm of the total assets of the enterprise at the end of the year. | Li et al., (2021) [16]; Lin et al., (2019) [25] | CSMAR |
| | Fixed asset ratio (Fixed) | The proportion of fixed assets to total assets. | Zhang, Rong and Ji (2019) [3] | CSMAR |
| | Gearing Ratio (Lev) | The proportion of total liabilities to total assets. | Jin and Xu (2020) [59]; Przychodzen and Przychodzen (2015) [1] | CSMAR |
| | Research and development investment (R&D) | The proportion of current R&D investment to operating revenue. | Przychodzen, Leyva-de la Hiz and Przychodzen (2020) [22] | CSMAR |
| | Government subsidies (Subsidy) | The logarithm of the amount of government subsidies received by the enterprise in the year. | Xie et al., (2016) [58] | CSMAR |
| | ISO14001 certification (ISO14001) | If the enterprise passes the ISO14001 audit in the year, the value is assigned as 1, otherwise it is 0. | de Paula et al., (2020) [24]; Przychodzen and Przychodzen (2015) [1] | CSMAR |

### 3.3. Model Section

Before conducting the panel data regression analyses, we first used the Hausman test to determine whether we should use a fixed or random effects model, and the results

suggested that we should use a fixed effects model. In order to try to avoid the regression results being biased by the omission of important explanatory variables, we used a two-way fixed-effects model that controls for individual ($\mu_{ind}$) and year ($\mu_t$) in the regression.

$$FP_{it} = \alpha_0 + \alpha_1 \times LnSubGI_{it} + \alpha_2 \times LnStrGI_{it} + \alpha_n \times Controls_{it} + \mu_{ind} + \mu_t + \varepsilon_{it} \quad (1)$$

$$FP_{it} = \beta_0 + \beta_1 \times LnSubGI_{it} + \beta_2 \times LnStrGI_{it} + \beta_n \times Controls_{it} + \mu_{ind} + \mu_t + \varepsilon_{it} \quad (2)$$

where $FP_{it}$ is the financial performance of a given company in a given year, measured as ROA; $EP_{it}$ is the environmental performance of a given company in a given year, measured as the carbon dioxide ($CO_2$) emissions per unit of assets; $LnSubGI_{it}$ is the substantive green innovation of a given company in a given year, measured as the number of green innovation patents obtained by a firm in a given year; $LnStrGI_{it}$ is the strategic green innovation of a given company in a given year, measured as the number of green utility model patents obtained by a firm in a given year; $Controls_{it}$ is the control variables, including enterprise size, fixed asset ratio, gearing ratio, research and development investment, government subsidies, and ISO14001 certification; $\mu_{ind}$ is a dummy variable representing a given firm; and $\mu_t$ is a dummy variable representing a given year of analysis.

## 4. Empirical Findings and Discussion

### 4.1. Descriptive Statistics

Table 2 reports the statistical results of the variables. It can be seen that the means of financial performance (FP) and environmental performance (EP) of the samples are 3.964 and 20.12, respectively, the median values are 3.869 and 17.69, respectively, and the standard deviations are 6.082 and 11.64, respectively, indicating a more even distribution of the dependent variables. In terms of green innovation, the distribution of green innovation among the sample companies is highly skewed. On average, firms received 3.637 substantive green patents and 9.958 strategic green patents per year during the sample period. The minimum and maximum values of substantive green patents are 0 and 533, respectively, with a median of 0. The minimum and maximum values of strategic green patents are 0 and 715, respectively, with a median of 3. This suggests that green innovations, especially substantive rate innovations, are still new to Chinese firms, which is consistent with the findings of Zhang et al. [3]. This also suggests a structural imbalance in green innovation activities in China, resulting in the number of high-quality substantive green innovations being much lower than the number of low-quality strategic green innovation, which is consistent with the findings of Zhao et al. [35]. The minimum value of SubGI and StrGI is 0, the maximum values are 533 and 715, and the standard deviations are 18.94 and 33.31, respectively, indicating a big difference between substantive green innovation and strategic green innovation in the samples. In addition, the correlation matrix in Table 3 shows that the substantive green innovation indicator LnSubGI positively correlates with FP, while the strategic green innovation indicator LnStrGI negatively correlates with FP. Similarly, it also shows that LnSubGI positively correlates with EP, and LnStrGI negatively correlates with EP. This preliminarily reveals the relationship between substantive green innovation and strategic green innovation and enterprises' financial and environmental performance.

### 4.2. Results of Panel Regression

Table 4 reports the results of two-way fixed-effects regressions of substantive and strategic green innovations on corporate financial and environmental performance. The table includes four columns, where columns (1) and (2) represent the baseline model with control variables. Column (3) displays the regression results of substantive and strategic green innovations on financial performance, while column (4) shows the regression results of substantive and strategic green innovations on environmental performance.

**Table 2.** Statistical description.

| Variable | Mean | Median | Skewness | SD | Min | Max |
|---|---|---|---|---|---|---|
| FP | 3.964 | 3.869 | −2.136 | 6.082 | −49.27 | 21.84 |
| EP | 20.12 | 17.69 | 1.729 | 11.64 | 1.949 | 82.47 |
| TotallGI | 13.59 | 4 | 11.75 | 47.22 | 1 | 1075 |
| SubGI | 3.637 | 0 | 15.28 | 18.94 | 0 | 533 |
| StrGI | 9.958 | 3 | 11.34 | 33.31 | 0 | 715 |
| LnSubGI | 0.660 | 0 | 1.661 | 0.919 | 0 | 4.898 |
| LnStrGI | 1.539 | 1.386 | 0.817 | 1.054 | 0 | 5.159 |
| Size | 22.56 | 22.35 | 0.672 | 1.368 | 19.77 | 26.46 |
| Fixed | 22.02 | 18.94 | 0.947 | 14.79 | 0.158 | 72.46 |
| Lev | 44.05 | 43.98 | 0.0817 | 19.26 | 3.131 | 92.46 |
| R&D | 4.617 | 3.830 | 2.403 | 3.916 | 0.0200 | 32.26 |
| Subsidy | 17.02 | 16.96 | −0.0282 | 1.518 | 10.82 | 20.75 |
| ISO14001 | 0.311 | 0 | 0.815 | 0.463 | 0 | 1 |

**Table 3.** Correlation matrix of the variables.

| Variable | FP | EP | LnSubGI | LnStrGI | Size | Fixed | Lev | R&D | Subsidy | ISO14001 |
|---|---|---|---|---|---|---|---|---|---|---|
| FP | 1 | | | | | | | | | |
| EP | 0.159 *** | 1 | | | | | | | | |
| LnSubGI | 0.022 ** | 0.037 *** | 1 | | | | | | | |
| LnStrGI | −0.076 *** | −0.004 | 0.424 *** | 1 | | | | | | |
| Size | 0.002 | 0.277 *** | 0.331 *** | 0.385 *** | 1 | | | | | |
| Fixed | −0.090 *** | 0.058 *** | −0.021 ** | −0.027 *** | 0.238 *** | 1 | | | | |
| Lev | −0.373 *** | 0.130 *** | 0.170 *** | 0.322 *** | 0.486 *** | 0.138 *** | 1 | | | |
| R&D | −0.026 *** | −0.267 *** | 0.013 | −0.071 *** | −0.276 *** | −0.256 *** | −0.326 *** | 1 | | |
| Subsidy | 0.023 ** | 0.079 *** | 0.337 *** | 0.369 *** | 0.649 *** | 0.118 *** | 0.332 *** | 0.012 | 1 | |
| ISO14001 | 0.046 *** | 0.063 *** | −0.008 | −0.011 | 0.025 *** | 0.009 | −0.023 ** | 0.019 ** | 0.005 | 1 |

Note: Table contains Pearson's correlation coefficient. * indicates $p < 0.1$, ** indicates $p < 0.05$, *** indicates $p < 0.01$.

**Table 4.** The impact of substantive and strategic green innovations on corporate financial and environmental performance.

| Variables | (1) FP | (2) FP | (3) EP | (4) EP |
|---|---|---|---|---|
| LnSubGI | | | 0.225 *** | −0.272 *** |
| | | | (0.075) | (0.096) |
| LnStrGI | | | −0.226 *** | −0.292 *** |
| | | | (0.072) | (0.092) |
| Size | 1.160 *** | 0.064 | 1.247 *** | 0.185 |
| | (0.152) | (0.195) | (0.154) | (0.198) |
| Fixed | −0.079 *** | 0.039 *** | −0.080 *** | 0.039 *** |
| | (0.008) | (0.010) | (0.008) | (0.010) |
| Lev | −0.197 *** | −0.006 | −0.196 *** | −0.006 |
| | (0.006) | (0.007) | (0.006) | (0.007) |
| R&D | −0.675 *** | −0.609 *** | −0.667 *** | −0.607 *** |
| | (0.029) | (0.037) | (0.029) | (0.037) |
| Subsidy | 0.136 ** | −0.551 *** | 0.163 *** | −0.530 *** |
| | (0.060) | (0.078) | (0.061) | (0.078) |
| ISO14001 | −0.103 | −0.341 * | −0.095 | −0.371 ** |
| | (0.141) | (0.181) | (0.141) | (0.181) |
| _cons | 5.829 *** | 31.427 *** | 4.825 *** | 30.722 *** |
| | (1.202) | (1.545) | (1.231) | (1.581) |
| Individual/Year | yes | yes | yes | yes |
| N | 10,940 | 10,940 | 10,940 | 10,940 |
| Adj-$R^2$ | 0.477 | 0.766 | 0.478 | 0.767 |

Note: Robust standard errors clustered on firm level in parentheses. * indicates $p < 0.1$, ** indicates $p < 0.05$, *** indicates $p < 0.01$.

As can be seen from columns (1) and (2), Size, Fixed, Lev, R&D, and Subsidy all have a significant effect on corporate financial performance, while Fixed, R&D, Subsidy, and ISO14001 all have a significant effect on corporate environmental performance.

The coefficients of substantive green innovation and strategic green innovation in column (3) are 0.225 and $-0.226$, respectively, and both are statistically significant at the 1% level. This shows a significant positive correlation between substantive green innovation (LnSubGI) and ROA, while a significant negative correlation exists between strategic green innovation (LnStrGI) and ROA. These findings suggest that a higher number of firms' substantive green innovations is more favorable to the firm's financial performance, while a higher number of strategic green innovations is more detrimental to the firm's financial performance, supporting H1a and H1b. The possible explanation for this is that substantive green innovation has a more significant effect on energy and material efficiency, which enhances the competitive differentiation of green products, leading to lower production costs and higher sales revenues, ultimately resulting in better financial performance. However, strategic green innovation not only has a minimal role in reducing costs and boosting sales through green production but also unilaterally increases firms' equipment acquisition and innovation costs, ultimately leading to a loss in firms' financial performance.

In column (4), the regression analysis shows that the coefficients of substantive green innovation (LnSubGI) and strategic green innovation (LnStrGI) are $-0.272$ and $-0.292$, respectively. Both coefficients are statistically significant at a 1% level. This implies that both substantive green innovation and strategic green innovation have the potential to reduce $CO_2$ emissions per unit of assets and improve the environmental performance of enterprises. This finding supports H2a and H2b. On this basis, we refer to Xie and Zhu [60] to standardize the regression coefficients of column (4) to compare the relative magnitude of the promotional effects of substantive green innovation and strategic green innovation on firms' environmental performance (for standardization, the untreated regression coefficients were multiplied by the standard deviation of that independent variable and then divided by the standard deviation of the dependent variable). After eliminating the effects of differences in magnitude and order of magnitude, the effects of substantive green innovation and strategic green innovation on environmental performance are $-0.021$ ($-0.272 \times 0.919/11.64 = -0.021$) and $-0.026$ ($-0.209 \times 1.054/11.64 = -0.026$), respectively. The results indicate that the extent of reduction in $CO_2$ emissions per unit of assets due to strategic green innovation is greater than that of substantive green innovation. Therefore, the enhancement effect of substantive green innovation on environmental performance is smaller than that of strategic green innovation. This validates H2c. The reason for this difference may be attributed to the fact that strategic green innovation can directly reduce the emissions of three wastes and, thus, have a more direct and rapid impact on environmental performance, whereas substantive green innovation can reduce the negative impacts of products on the environment during production, use and recycling, which is more indirect.

*4.3. Robustness Tests*

In order to ensure the reliability of the conclusions of this paper, we performed robustness tests in four ways.

4.3.1. Controlling for Previous Period Performance

To eliminate the effect of the previous period's performance on the current period, we referred to a previous study [42] and added the previous period's financial performance and environmental performance as control variables, respectively. The estimation results are shown in columns (1)–(2) of Table 5.

**Table 5.** Robustness test.

| Variables | (1)<br>FP | (2)<br>EP | (3)<br>FP1 | (4)<br>EP1 | (5)<br>FP | (6)<br>EP | (7)<br>FP | (8)<br>EP |
|---|---|---|---|---|---|---|---|---|
| LnSubGI | 0.225 *** | −0.224 ** | 0.629 *** | −0.009 ** | 0.185 ** | −0.214 ** | 0.300 *** | −0.201 * |
| | (0.086) | (0.100) | (0.202) | (0.004) | (0.083) | (0.104) | (0.098) | (0.114) |
| LnStrGI | −0.181 ** | −0.252 ** | −0.338 * | −0.015 *** | −0.208 *** | −0.254 ** | −0.302 *** | −0.253 ** |
| | (0.086) | (0.103) | (0.195) | (0.004) | (0.081) | (0.101) | (0.089) | (0.125) |
| $FP_{t-1}$ | 0.038 *** | | | | | | | |
| | (0.014) | | | | | | | |
| $EP_{t-1}$ | | 0.332 *** | | | | | | |
| | | (0.012) | | | | | | |
| Size | 1.231 *** | −0.205 | 4.802 *** | 0.055 *** | 1.040 *** | 0.561 *** | 1.217 *** | −0.811 *** |
| | (0.214) | (0.243) | (0.419) | (0.009) | (0.174) | (0.217) | (0.198) | (0.253) |
| Fixed | −0.093 *** | 0.056 *** | −0.139 *** | 0.002 *** | −0.095 *** | 0.033 *** | −0.081 *** | 0.024 * |
| | (0.011) | (0.013) | (0.022) | (0.000) | (0.009) | (0.011) | (0.010) | (0.013) |
| Lev | −0.208 *** | −0.016 * | −0.439 *** | −0.001 *** | −0.189 *** | −0.007 | −0.197 *** | −0.002 |
| | (0.008) | (0.009) | (0.016) | (0.000) | (0.006) | (0.008) | (0.007) | (0.009) |
| R&D | −0.726 *** | −0.559 *** | −1.438 *** | −0.040 *** | −0.663 *** | −0.582 *** | −0.674 *** | −0.618 *** |
| | (0.040) | (0.045) | (0.079) | (0.002) | (0.031) | (0.038) | (0.037) | (0.048) |
| Subsidy | 0.286 *** | −0.306 *** | 0.265 | −0.020 *** | 0.234 *** | −0.437 *** | 0.157 ** | −0.448 *** |
| | (0.078) | (0.089) | (0.165) | (0.004) | (0.071) | (0.088) | (0.072) | (0.091) |
| ISO14001 | −0.107 | −0.493 ** | −0.129 | −0.021 *** | −0.074 | −0.489 ** | 0.136 | −0.304 |
| | (0.175) | (0.198) | (0.382) | (0.008) | (0.154) | (0.193) | (0.176) | (0.224) |
| _cons | 3.696 ** | 23.126 *** | −7.906 ** | 2.957 *** | 5.449 *** | 26.809 *** | 5.086 *** | 37.402 *** |
| | (1.747) | (1.997) | (3.331) | (0.073) | (1.367) | (1.706) | (1.615) | (2.061) |
| Individual/Year | yes | yes | yes | yes | yes | yes | yes | yes |
| N | 7388 | 7388 | 10,934 | 10,940 | 9323 | 9323 | 8215 | 8215 |
| Adj-R$^2$ | 0.495 | 0.813 | 0.303 | 0.784 | 0.481 | 0.767 | 0.597 | 0.841 |

Note: Robust standard errors clustered on firm-level in parentheses. * indicates $p < 0.1$, ** indicates $p < 0.05$, *** indicates $p < 0.01$.

### 4.3.2. Different Proxy Variables

In order to ensure the reliability of the findings, we drew on existing studies [61,62] and replaced the dependent variables, changing FP from return on assets (ROA) to return on equity (ROE) and changing EP from carbon dioxide emissions per unit of assets to carbon dioxide emissions per capita of the enterprise, to obtain FP1 and EP1. The results of the re-examination are shown in columns (3)–(4) of Table 5.

### 4.3.3. Retaining the Manufacturing Sample

Manufacturing enterprises are the core part of the real economy, the main contributor to environmental pollution, as well as an essential area for green innovation to exert value. To ensure the reliability of the conclusions, we narrowed the sample to include only manufacturing enterprises. The number of observations after narrowing is 9323, and the results are shown in columns (5)–(6) of Table 5.

### 4.3.4. Excluding the Interference Sample

Due to the continued impact of the COVID-19 outbreak and the government's strict epidemic prevention and control policies, many Chinese companies were in an intermittent shutdown from 2020 to 2022. To eliminate the influence of the COVID-19 pandemic, we excluded the 2020–2021 samples and obtained 8215 observations. The results are presented in columns (7)–(8) of Table 5.

As can be seen in Table 5, substantive green innovation still contributes significantly to firms' financial performance and environmental performance, while strategic green innovation significantly harms firms' financial performance but contributes to environmental performance. Furthermore, it was observed that substantive green innovation contributes less to environmental performance compared to strategic green innovation (the

standardized regression coefficients for substantive green innovations on environmental performance are smaller than those for strategic green innovations). These findings indicate that the results of this study are reliable and valid.

### 4.4. Heterogeneity Analysis

State-owned enterprises (SOEs) and non-state-owned enterprises (NSOEs) have different social statuses and positions in the national economic system. Compared with non-state-owned enterprises, SOEs bear more responsibilities and social obligations. Therefore, the ownership of enterprises may affect the "Porter hypothesis". In addition, the "Porter hypothesis" may also vary depending on the level of governmental environmental concerns in the region where the firms are located. To further examine hypotheses 1–2, this paper analyzes the heterogeneity in terms of firm ownership and regional government environmental concerns.

### 4.4.1. The Heterogeneity of Firm Ownership

In order to investigate the effects of firm ownership on the relationship between substantive and strategic green innovations and firm performance, we added SOE and its interaction term with substantive green innovation and strategic green innovation to the original regression analysis. SOE is an indicator of enterprise ownership type, with SOEs given a value of 1 and non-SOEs given 0. The results are presented in columns (1)–(2) of Table 6.

In column (1) of Table 6, we observe that the estimated coefficient of LnSubGI × SOE on FP is −0.505, which is statistically significant at the 1% level. Additionally, the estimated coefficient of LnStrGI × SOE on FP is 0.240, which is statistically significant at the 10% level as well. These results suggest that in SOEs, the positive impact of substantive green innovation on financial performance is weaker, while the negative impact of strategic green innovation on financial performance is also weaker. This may be due to the fact that SOEs have a closer relationship with the government than non-SOEs, and SOEs are not only concerned with profit maximization but also have both the responsibility and obligation to create social welfare value [63]. Therefore, the green innovation activities of SOEs are aimed at responding to the government's environmental policies, ensuring their effective implementation. However, SOEs tend to take a more moderate approach to green innovation, neither pursuing top-end green technologies excessively nor low-end green technologies excessively. This moderation results in SOEs moderating the impact of substantive and strategic green innovations on financial performance.

### 4.4.2. The Heterogeneity of Government Environmental Concerns

A question worth exploring is whether government environmental concern, as an informal environmental regulation tool of the government, affects the relationship between green innovation and firm performance. Government environmental concerns (GEC) are an indicator of the intensity of environmental concern of the government where the enterprise is located. With reference to Chen et al. [54], we used textual analyses to measure GEC. Based on the text information of the government work report at each province level, we searched and identified keywords in the text with ecological and environmental feature words, such as "environmental protection and governance", "environmental governance measures", "energy consumption", and "environmental regulation", and then counted the ratio of the word frequency of these words in the text of the government work report of each province to the word count of the whole government work report as a proxy for GEC. When the GEC of the province where the firm is located is above the mean, it takes the value of 1, and when it is below the mean, it takes the value of 0. The results are shown in columns (3)–(4) of Table 6.

**Table 6.** Heterogeneity analysis.

| Variables | (1) FP | (2) EP | (3) FP | (4) EP |
|---|---|---|---|---|
| LnSubGI | 0.437 *** | −0.058 | 0.617 ** | 0.151 |
| | (0.098) | (0.126) | (0.249) | (0.320) |
| LnStrGI | −0.313 *** | −0.165 | −0.130 | 0.198 |
| | (0.090) | (0.115) | (0.216) | (0.277) |
| LnSubGI × SOE | −0.505 *** | −0.468 ** | | |
| | (0.147) | (0.189) | | |
| LnStrGI × SOE | 0.240 * | −0.306 * | | |
| | (0.138) | (0.177) | | |
| LnSubGI × GEC | | | −0.440 * | −0.351 |
| | | | (0.248) | (0.318) |
| LnStrGI × GEC | | | −0.097 | −0.532 * |
| | | | (0.217) | (0.279) |
| SOE | −0.871 * | 1.167 * | | |
| | (0.473) | (0.608) | | |
| GEC | | | 1.197 *** | −0.136 |
| | | | (0.394) | (0.506) |
| Size | 1.261 *** | 0.182 | 1.235 *** | 0.196 |
| | (0.154) | (0.198) | (0.155) | (0.198) |
| Fixed | −0.080 *** | 0.038 *** | −0.081 *** | 0.040 *** |
| | (0.008) | (0.010) | (0.008) | (0.010) |
| Lev | −0.195 *** | −0.007 | −0.195 *** | −0.006 |
| | (0.006) | (0.007) | (0.006) | (0.007) |
| R&D | −0.668 *** | −0.611 *** | −0.661 *** | −0.611 *** |
| | (0.029) | (0.037) | (0.029) | (0.037) |
| Subsidy | 0.168 *** | −0.528 *** | 0.174 *** | −0.552 *** |
| | (0.061) | (0.078) | (0.061) | (0.078) |
| ISO14001 | −0.081 | −0.382 ** | −0.090 | −0.360 ** |
| | (0.141) | (0.181) | (0.141) | (0.180) |
| _cons | 4.903 *** | 30.254 *** | 3.593 *** | 30.964 *** |
| | (1.231) | (1.581) | (1.280) | (1.643) |
| Individual/Year | yes | yes | yes | yes |
| N | 10,940 | 10,940 | 10,936 | 10,936 |
| Adj-$R^2$ | 0.500 | 0.770 | 0.573 | 0.809 |

Note: Robust standard errors clustered on firm-level in parentheses. * indicates $p < 0.1$, ** indicates $p < 0.05$, *** indicates $p < 0.01$.

As can be seen in column (3) of Table 6, the contribution of substantive green innovation to financial performance is weaker for firms in regions with high government environmental concerns. This may be due to the fact that higher government environmental concerns result in firms not being proactive but forced to invest large amounts of corporate capital in green innovation and related unproductive activities, creating barriers to profitability.

## 5. Research Conclusions, Insights and Limitations

### 5.1. Conclusions

Based on the panel data of listed companies in China from 2010 to 2021, this study originally investigates the impact of substantive green innovation and strategic green innovation on corporate financial performance and environmental performance. We find that substantive green innovation has both economic and environmental benefits, fully supporting the "Porter hypothesis". However, strategic green innovation distorts the "Porter hypothesis", i.e., although it can improve environmental performance, it hurts firms' economic rewards. Furthermore, the heterogeneity analyses show that the financial performance enhancement effect of substantive green innovation is weaker in state-owned enterprises (SOEs) and in regions with high government environmental concerns; similarly, the financial performance detrimental effect of strategic green innovation is also weaker in SOEs.

The findings of this paper explain, to some extent, why firms invest in such "costly" strategic green innovations in large numbers, as their existence is justified. Under the government's formal and informal environmental regulation policies, low-cost, low-threshold, and low-quality strategic green innovations can bring significant and rapid environmental performance improvements to firms in the face of environmental pressures from regulatory stakeholders in the short term and can ensure the legitimacy of firms. In addition, whereas substantive green innovation requires a high overall investment in R&D, strategic green innovation allows firms to build up green knowledge and capabilities as a means of transition to substantive green innovation. Overall, strategic green innovation primarily exists for the sake of firm legitimacy, while substantive green innovation can achieve both efficiency and legitimacy outcomes. Our findings suggest that it is worthwhile for Chinese firms to "go green", and that developing substantive green innovation is a viable strategy for firms to achieve high profits and high environmental performance.

Our research is original and responds to the call from scholars [14] for research on the impact of green innovation strategies on firm performance. This is because previous academic studies on this relationship have not reached consistent conclusions. While some previous studies suggest that green innovation can positively impact firms' financial performance [19], others argue that it brings additional costs and harms firms' financial performance [64]. Recent studies have pointed out that the type of green innovation is a prerequisite for firm performance [23]. By building on this line of thinking and focusing on the motivational and qualitative types of green innovations, our analysis reveals that the financial impact of green innovations depends on whether firms implement substantive or strategic green innovations. Furthermore, our findings suggest that both substantive and strategic green innovations can positively impact firms' environmental performance, which is consistent with previous studies that suggest that green innovations can lead to positive environmental performance for firms [13]. Additionally, we find that strategic green innovation is more advantageous in terms of its contribution to environmental performance. The findings of our paper enrich the literature in the field of green innovation and guide firms to actively implement green technology innovation practices.

### 5.2. Management and Practical Insights

This study provides two management insights. First, green innovation activities not only impact a company's cost and profitability but also the natural environment on which we all depend. Therefore, firm managers should adopt a positive attitude toward sustainability, take proactive risks, and invest in green technology innovation activities to achieve a win–win situation for the environment, society, and the firm. Second, enterprises should pay extra attention to improving the quality of green innovations because only high-quality substantive green innovations can truly reduce production costs, sustainably improve product differentiation, and build multiple competencies of enterprises, which in turn will lead to improvements in both financial and environmental performance.

The implementation of green innovation activities by enterprises is beneficial to the enterprises themselves, to the environment, and even to society. Governments should take steps to promote green innovation, specifically substantive green innovation, among businesses. First, governments can encourage and urge enterprises to engage in green innovation activities through the establishment of appropriate environmental laws and regulations to effectively address the environmental needs of society and realize the sustainability of the ecology. Second, governments should implement the concept of "substantive green innovation to promote substantive sustainable development" and encourage and guide enterprises to engage in substantive green innovation. For example, governments should take the lead in establishing a green technology platform to solve common technical problems and lower the cost and threshold for enterprises to carry out substantive green innovation. In addition, governments can fully implement this concept into the formulation of green innovation incentive policies by subsidizing the price of green products and providing tax rebates for green innovation projects with high technological content to

achieve the goals of continuous emission reduction and sustainable environmental management by enterprises. Third, studies have shown that only financially unconstrained firms can achieve higher economic returns from environmental investments [55]. As developing green innovations require significant costs, they may cause firms to be financially constrained, or firms' budgetary constraints may constrain the implementation of green innovations by firms. Therefore, in addition to mandatory policies such as environmental regulations, governments need to consider providing a favorable financial environment for firms to increase their financial flexibility in order to promote the incorporation of green technological innovations into their corporate strategies and, thus, increase their contribution to environmental protection.

*5.3. Limitations and Future Research*

There are several possible limitations of this study. First, our sample is limited to listed companies in China. Therefore, future research is necessary to understand to what extent the findings and conclusions of this paper apply to companies in other countries. Second, while our findings show a significant difference in the impact of substantive green innovation and strategic green innovation on corporate financial performance and environmental performance, we do not delve into the specific mechanisms by which this difference in impact is formed. Third, since there is no unanimity in the academic community on the measurement of environmental performance, and because of the feasibility of measuring panel data, we measure environmental performance based on firms' $CO_2$ emissions. Although this is the best measurement we can perform within our capacity, it may not be a comprehensive measure of environmental performance. Future research can try to broaden the scope of environmental performance measurement. Fourth, this study only focuses on green technological innovations that can be measured using green patents. It is necessary to include other types of green innovations in future studies, such as marketing, organizational, and logistic innovations. Fifth, this study only analyzed two factors that may affect the relationship between green innovation and firm performance. Future research may be able to explore better whether specific firm characteristics and environmental conditions affect the relationship between green innovation and firm performance, such as the green awareness of managers, the public's concern for the environment, and the degree of regional intellectual property rights protection, which may be effective in moderating the relationship between the two. In conclusion, it is hoped that this study will inspire more scholars to explore the relationship between green innovation and firm performance to promote the vigorous development of research in the field of green innovation.

**Author Contributions:** The authors worked together for this research, but, per structure, the contributions are outlined as follows: conceptualization M.L. and L.L.; methodology, software validation and resources, L.L. and A.F.; data analysis, L.L.; writing—original draft preparation M.L. and L.L.; writing—review and editing, M.L., L.L. and A.F. All authors have read and agreed to the published version of the manuscript.

**Funding:** This research received no external funding.

**Institutional Review Board Statement:** Not applicable.

**Informed Consent Statement:** Not applicable.

**Data Availability Statement:** The data presented in this study are available on request from the corresponding author.

**Acknowledgments:** The authors thank the editor and anonymous reviewers for their useful comments and suggestions.

**Conflicts of Interest:** The authors declare no conflicts of interest.

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
