# Peer review of "The Impact of Green Innovation on Corporate Performance: An Analysis Based on Substantive and Strategic Green Innovations"

_sustainability, doi:10.3390/su16062588_

Round 1
Reviewer 1 Report
Comments and Suggestions for Authors
This study aims to explore the effects of green innovation on corporate performance, working on a sample of Chinese companies between 2010 and 2021.
The introduction adequately sets the general background for the research. Additionally, the authors emphasize the main contributions of the study, which are related to analyzing the impact of green innovations on corporate financial and environmental performance in the case of Chinese companies, finding pieces of evidence of the effects of green innovation on companies’ performance; finding shreds of evidence showing a relationship between ownership and government environmental concerns and financial performance; finding evidence supporting policies for promoting green innovation.
The literature is adequate for the topic and relevant for the field. However, it is advisable to expand it to discuss other recent relevant studies, such as:
https://doi.org/10.3390/su142316159
https://doi.org/10.1007/s11356-023-27796-3
Please insert a table presenting the variables and their sources. In addition, please comprehensively explain the rationale behind selecting the model.
Please discuss the main results in the context of previous research results, emphasizing the original findings of this study or the support to be found in previous literature for the results.
The conclusions enhance the value of the study reprising the main findings and presenting the limitations of the research and future developments. Besides the management implications, hinting at some macroeconomic implications or policy recommendations is advisable.
Author Response
Dear Reviewer,
We would like to express our sincere gratitude for the valuable feedback you provided on our manuscript (sustainability-2770453). Your comments, suggestions, and corrections were insightful and helped us improve the quality of the manuscript. The revised version of the manuscript highlights the changes made in red.
Please see the attachment.

Reviewer 2 Report
Comments and Suggestions for Authors
Article is exploring a relevant topic. There are many studies that explore the same topic, but I consider that approach and how research was perfomed is interesting. I have some suggestions to improve article. First is to include methodology in the abstract and also in introduction. Second, in the paragraph that is in lines 50-59, I think that is important to refer sources. Third, I think that theoretical review is too short and should be expanded.
Author Response

(The authors gave the same response as above.)

Reviewer 3 Report
Comments and Suggestions for Authors
The English is very good, so only one minor revision should be done
Author Response

(The authors gave the same response as above.)

Round 2
Reviewer 3 Report
Comments and Suggestions for Authors
Thank you for accepting my comments and recommendation and for your efforts to improve your work. The changes that you have made highly improved the final document and correctly undressed my main concerned and suggestions.
Author Response
Dear Reviewer,
We want to express our appreciation for your feedback and for taking the time to review our revised manuscript. Thank you for recognizing the efforts we put into it.
Best regards.